# Serum Levels of CXCR4, SDF-1, MCP-1, NF-κB and ERK1/2 in Patients with Skeletal Fluorosis

**DOI:** 10.3390/ijerph192416555

**Published:** 2022-12-09

**Authors:** Yaqian Zhao, Guanglan Pu, Yanan Li, Hong Jiang, Qiang Zhang, Ping Chen, Qing Lu, Mingjun Wang, Rui Yang

**Affiliations:** 1Department of Public Health, Medical College, Qinghai University, Xi’ning 810016, China; 2Department of Endemic Disease Prevention and Control, Qinghai Institute for Endemic Disease Prevention and Control, Xi’ning 811602, China

**Keywords:** skeletal fluorosis, chemokines, extracellular signal-regulated kinase 1/2, nuclear factor-κB

## Abstract

C-X-C motif chemokine receptor 4 (CXCR4), stromal cell-derived factor-1 (SDF-1), monocyte chemoattractant protein-1 (MCP-1), extracellular signal-regulated kinase 1/2 (ERK1/2) and nuclear factor-κB (NF-κB) affect bone cells and play an important role in bone and joint diseases, but the data on CXCR4, SDF-1, MCP-1, ERK1/2 and NF-κB in the serum of skeletal fluorosis (SF) patients are inconclusive. Thus, according to the “Diagnostic Criteria for Endemic Skeletal Fluorosis” (WS 192-2008), we enrolled patients with SF (*n* = 60) as the SF group and those without SF as the controls (*n* = 60). Serum levels of CXCR4, SDF-1, MCP-1, ERK1/2 and NF-κB were detected by enzyme-linked immunosorbent assays (ELISAs). Serum SDF-1, CXCR4, MCP-1 and NF-κB levels were significantly higher in the SF group than in the control group. Within the serum of SF patients, CXCR4 and SDF-1 levels were positively correlated with NF-κB levels. There was no correlation between MCP-1 levels and those of ERK1/2 or NF-κB. SDF-1 and CXCR4 may activate the NF-κB pathway, and MCP-1 affects the occurrence and development of SF by regulating osteocytes through other pathways. The SDF-1/CXCR4 axis and MCP-1 signalling pathway provide a new theoretical basis for the occurrence and development of SF.

## 1. Introduction

Fluorine is widely present in the natural environment and is an indispensable trace element for human growth and development. An appropriate concentration of fluorine is beneficial to oral and bone health, while a high fluorine concentration has adverse effects on the body and causes fluorosis [1]. Endemic fluorosis, also known as geofluorosis, is present on all continents, affects a large population and is a serious public health problem [2]. According to fluorine source and intake route, endemic fluorosis is classified as drinking water-type, coal burning-type and brick tea-type [3].

Bone functions as the buffer for fluoride and the main fluoride reservoir in the body. Fluoride can affect osteoblasts (OBs) and osteoclasts (OCs), and exposure to a small amount of fluoride can promote OB proliferation and inhibit bone absorption by OCs [4]. Exposure to excessive fluoride can induce the destruction of OBs and the formation of OCs, leading to skeletal fluorosis (SF) [2,5]. Once this disease develops, it causes permanent changes [6]. Clinical manifestations of SF include osteoporosis (OP), osteosclerosis, and disabling deformity; thus, SF is considered one of the most serious forms of chronic fluorosis [7]. X-ray imaging is an important method for the diagnosis of SF. The main concerns related to this disease include the degree of osteosclerosis, osteomalacia, bone transformation, and periosseous and joint changes. The key factor is the degree of ossification of the interosseous membrane of the forearm, tibiofibular interosseous membrane, pronator teres muscle and soleus tendon. According to the imaging findings, SF can be classified into three grades, mild, moderate and severe, that reflect the severity of bone lesions.

At present, the pathogenesis of SF is not well understood. In recent years, research on the mechanism underlying the occurrence and development of SF has mainly focused on the regulation of bone turnover by various cells [8], and extensive insight into fluorine-induced oxidative stress, apoptosis and signalling pathways has been reported [2]. Several investigations have demonstrated that fluoride can induce oxidative stress and modulate intracellular redox homeostasis, lipid peroxidation and protein carbonyl content, as well as cause apoptosis and alter gene expression, such as the stress response, metabolic enzymes, the cell cycle, cell–cell communications and signal transduction genes [2]. Fluoride induces apoptotic cell death by disturbing signalling messages through multiple mechanisms. The signalling pathway of the apoptotic response to fluoride exposure involves Bcl-2, Bax, G proteins and Fas [2,9,10]. Research data also suggest that fluoride influences synthesis and that it influences distinct signalling pathways involved in proliferation and apoptosis, including the p53, activator protein-1 (AP-1), nuclear factor kappa B (NF-κB) and mitogen-activated protein kinase (MAPK) pathways [11,12,13]. Scholars believe that the mechanism of action of fluorine and fluoro-aluminium compounds on bone is mainly realised through the MAPK signalling pathway [9].

Stromal cell-derived factor-1 (SDF-1) activates all three MAPK signalling pathways, including the extracellular signal-regulated kinase 1/2 (ERK1/2) pathway, c-Jun N-terminal kinase (JNK) pathway and p38 pathway [14]. SDF-1, also known as C-X-C motif chemokine ligand 12 (CXCL12), is a member of the chemokine family. Chemokines play important roles in normal bone physiology and bone diseases. CC and CXC chemokines can stimulate new bone formation and old bone remodelling and play important roles in many bone-related pathologies [15]. Studies have shown that monocyte chemoattractant protein-1 (MCP-1), also known as C-C motif chemokine ligand 2 (CCL2), CCL3, CCL4 and SDF-1 play important roles in maintaining the balance between bone resorption and bone formation [16]. MCP-1 and SDF-1 are widely studied chemokines in bone-related diseases [17,18]. SDF-1, also known as C-X-C motif chemokine ligand 12 (CXCL12), is an agonist of the G protein-coupled receptor (GPCR) C-X-C motif chemokine receptor 4 (CXCR4) [19,20]. The SDF-1/CXCR4 axis has been thoroughly studied; the two components of this chemokine/receptor pair are widely expressed. The SDF-1/CXCR4 axis influences angiogenesis and leukocytes and is involved in various pathologies, including cancer and autoimmune and inflammatory disorders [21]. SDF-1 and CXCR4 are essential for the recruitment, localization, maintenance, development and differentiation of progenitor stem cells of the skeletal system and are related to the development and function of mature and precursor OCs, OBs and other cells [22]. The SDF-1/CXCR4 axis is also involved in OB differentiation, OC formation and endochondral ossification [22,23,24]. CXCR4 can cooperate with bone morphogenetic protein (BMP) signal transduction to regulate OB differentiation [22]. CXCR4 is highly expressed in pre-OCs. After SDF-1 interacts with CXCR4, pre-OCs are recruited into specific regions of bone marrow, wherein receptor activator of nuclear factor κB ligand (RANKL) promotes OC development [22]; CXCR4 expression decreases during RANKL-mediated OC formation [24]. Hypertrophic chondrocytes play an important role in endochondral ossification [23]. Studies have found that the SDF-1/CXCR4 signalling pathway can induce chondrocyte proliferation and hypertrophy in a mechanism regulated by a Runx2-mediated positive feedback loop. Chondrocyte proliferation and hypertrophy are processes of endochondral ossification. MCP-1 is a cytokine in the CC chemokine family that coordinates the migration of multiple cell types through C-C motif chemokine receptor 2 (CCR2) under physiological and pathological conditions, such as rheumatoid arthritis (RA) and other autoimmune diseases [25]. Studies have shown that MCP-1 promotes the migration of OC precursors, and the development of OCs and bone resorption are involved in bone degradation and regulate bone remodelling [26,27]. In OC progenitors, MCP-1 binds CCR2 to stimulate NF-κB and ERK1/2 signalling to increase RANKL expression and promote the differentiation to mature OCs [28]. In MCP-1/CCR2-deficient mice, bone resorption is reduced, OC formation and function are defective, and the number of OCs is reduced [29]. MCP-1 also affects OB function in bone remodelling [30]. MCP-1 affects the synthesis and metabolism of parathyroid hormone (PTH), which can increase the number of bone cells and regulate bone remodelling. In addition, PTH induces MCP-1 to promote the recruitment of bone precursors to remodelling sites and bone remodelling itself [30]. MCP-1 is also involved in the fusion of OBs with mature OCs [30].

SDF-1 and MCP-1 can stimulate ERK1/2 signalling, which are members of the MAPK family of signalling molecules [14]. The ERK1/2 signalling pathway regulates a variety of cellular responses, including proliferation, growth, differentiation, migration, survival, metabolism and transcription, and participates in many diseases, such as chronic inflammation, arthritis and OP [31]. Studies have found that ERK1/2 play key roles in regulating the function of OBs and OCs. ERK1/2 affect OB activation and differentiation. Core binding factor α1 (CBFα1) regulates OB differentiation and bone formation. ERK1/2 affects OBs by increasing CBFα1 expression on OB surfaces, resulting in increased bone metabolism and further bone sclerosis [32,33]. ERK1/2 are essential regulators of OC differentiation and apoptosis; these kinases promote the differentiation of bone marrow-derived macrophages into OCs by inducing the expression of RANKL and receptor activator of nuclear factor-κB (RANK) on the OC surface.

The SDF-1/CXCR4 axis may be an upstream regulator of the NF-κB signalling pathway [34]. Stimulation of ERK1/2 impacts the NF-κB signalling pathway through the following cascade: SDF-1/CXCR4→PI3→MAPK (RAS-RAF-MEK1/2-ERK1/2)→NF-κB [35]. NF-κB, a member of the nuclear transcription factor family, has a wide range of functions and is involved in embryonic development, cell proliferation and apoptosis, the immune response, the inflammatory response and other processes [36]. In many animal models, NF-κB family transcription factors have been shown to affect bone formation and to play important regulatory roles in the occurrence of bone-related diseases. NF-κB influences bone development and remodelling by regulating biological processes such as the proliferation, differentiation and apoptosis of OBs, OCs, osteocytes and chondrocytes, mainly in the direction of inhibition [36,37]. Recent studies found that activation of the classical NF-κB signalling pathway can decrease the differentiation ability of OBs, which can be enhanced by inhibiting the expression of NF-κB-p65 (p65). In addition, p65 can interact with Smad1 and Smad5 to inhibit NF-κB signalling [36]. NF-κB is involved in OC formation and activity and is essential for the differentiation of OC precursors into OCs [38]. Similar to the mechanism by which ERK1/2 promote OC differentiation, the NF-κB signalling pathway mediates the binding of RANKL to its cognate receptor RANK to induce OC formation [39].

SDF-1, CXCR4, MCP-1, ERK1/2 and NF-κB play important roles in maintaining the stability of bone, but there are very few studies on these factors in SF. Their role in SF remains unclear. Here, the serum levels of SDF-1, CXCR4, MCP-1, ERK1/2 and NF-κB were determined in SF patients, and the mechanism by which these proteins interact in SF was explored, providing novel evidence of the SDF-1/CXCR4 axis and MCP-1 signalling pathways underlying the occurrence and development of SF.

## 2. Materials and Methods

### 2.1. Study Participants

According to the “Diagnostic Criteria for Endemic Skeletal Fluorosis” (WS 192 -- 2008), which are mainly based on X-ray images, 60 adult SF patients in Zhiduo County, Yushu Prefecture, and Gangcha County, Haibei Prefecture, Qinghai Province, were included as the SF group, and 60 adults without SF were included in the same population as the control group. All patients in the SF group came from areas with endemic brick tea-type fluorosis. Patients with bone and joint diseases, arthritis, chronic inflammation, gout, cancer or a strong family history of joint cancer were excluded based on previous medical history inquiry, epidemiological investigation, clinical manifestations or laboratory tests. This research was supervised by the Medical Ethics Committee of Qinghai Institute of Endemic Disease Prevention and Control (2021SQCJ7901). The research participants’ right to know and right to choose were respected, informed consent forms were prepared, written informed consent was obtained and the study was carried out with the consent of the participants.

### 2.2. Sample Collection and Processing

Approximately 3~5 mL of fasting venous blood was collected on site and centrifuged at 3000× *g* r/min for 10 min, and the upper layer of serum was removed and placed in a centrifuge tube. The serum was transported at 4 °C and frozen at −80 °C.

### 2.3. Measurement of Cytokines

CXCR4 (Ronghui Innovation Technology Co., Ltd., Lot: 202111, Gansu, China), MCP-1 (Abcam, Lot: GR3423784-1, UK), SDF-1, ERK1/2 and NF-κB (Jiangsu Enzyme Labelling Biotechnology Co., Ltd., Lot: 202207, Jiangsu, China) levels in blood serum were assessed by enzyme-linked immunosorbent assays (ELISAs) with a wide detection range: 0–4 ng/mL (CXCR4), 0–300 pg/mL (MCP-1), 0–8 ng/mL (SDF-1), 0–4000 pg/mL (ERK1/2) and 0–80 ng/mL (NF-κB), respectively.

Absorbance values were detected at 450 nm using a Multiskan GO microplate reader (Thermo Scientific, CO, Waltham, MA, USA).

### 2.4. Statistical Analysis

Excel was used to generate a database, and SPSS 25.0 was used for data analysis. Normally distributed measurement data with homogeneous variance are presented as the mean ± SD, and an independent t-test was used to compare two groups. Data not conforming to a normal distribution are presented as the median and interquartile range [M (*P*_25_, *P*_75_)]; the Mann–Whitney U test was used to compare two groups, and the Kruskal–Wallis test was used for multigroup comparisons. Count data are presented as the percentage (%), and the *χ** test was used to analyse categorical data. Correlation analysis results are presented as the Spearman correlation coefficient. A *p*-value < 0.05 was considered to indicate statistical significance. The figures in this manuscript were generated using GraphPad Prism 8 software (GraphPad Software, San Diego, CA, USA).

## 3. Results

### 3.1. Demographic Data

Sixty participants each were enrolled in the SF group and the control group. Patients with SF were divided into three subgroups: 36 patients were assigned to the mild degeneration group, 20 to the moderate degeneration group and 4 to the severe degeneration group (Table 1).

### 3.2. X-ray Imaging Analysis of SF Patients

#### 3.2.1. Case Analysis of a Patient Diagnosed by X-ray with Mild SF

The patient is a 52-year-old female with persistent pain in the limbs and joints at rest who can engage in normal physical labour. The anteroposterior actinogram of the upper limb (Figure 1a) shows ossification of the interosseous membrane of the radius and ulna, with a cotton wadding appearance. Periarticular tendon and pronator teres muscle ossification, and degenerative changes in the elbow joint can be seen. The anteroposterior actinogram of both lower limbs (Figure 1b) shows ossification of the tibiofibular interosseous membrane, the shadow of the colliculus bony protrusion at the tibial and fibular and ossification of the knee joint capsule and tibial collateral ligament.

#### 3.2.2. Case Analysis of a Patient Diagnosed by X-ray with Moderate SF

The patient is a 49-year-old female with pain in the limbs, neck and waist and limited ability to exercise with a variable impact on labour participation. The anteroposterior actinogram of the upper limb (Figure 1c) shows ossification of the interosseous membrane of the radius and ulna, with a fin-like appearance, degeneration of the elbow joint, ossification of the pronator teres muscle and bony protrusions on the inner side of the ulna. The anteroposterior actinogram of both lower limbs (Figure 1d) shows ossification of the tibiofibular interosseous membrane, a lacelike bony protrusion shadow at the tibial and fibular interosseous membrane, ossification of the long peroneal muscle, knee joint capsule and tibial collateral ligament.

#### 3.2.3. Case Analysis of a Patient Diagnosed by X-ray with Severe SF

The patient is a 71-year-old male with severe bone and joint pain, limited activity and no ability to work. The anteroposterior actinogram of the upper limb (Figure 1e) shows obvious ossification of the interosseous membrane of the ulna and radius, forming a large fin-like ossification near the ulna and radius. The pronator teres muscle tendon shows obvious ossification, and a stalactite ossification shadow can be seen in the radial head. There is obvious degeneration of the elbow joint. The anteroposterior actinogram of both lower limbs (Figure 1f) shows ossification of the tibiofibular interosseous membrane, with a colliculus appearance. The peroneus longus tendons show obvious ossification, as do the soleus tendon, knee joint capsule and tibial collateral ligament.

Ossification of the interosseous membrane has a high incidence and specificity in SF and is one of the important diagnostic signs of SF. Ossification of the interosseous membrane was more obvious in patients with severe SF than in those with mild or moderate SF.

### 3.3. X-ray Grade Analysis of Ages, Age Groups and Sexes

X-ray diagnosis results were performed for three different aspects, including correlations with age, age group and sex. There was a positive correlation between age and X-ray diagnosis grade (Figure 2). There was no significant difference in the X-ray diagnosis of mild, moderate or severe SF among patients of different sexes (*χ** = 2.18, *p >* 0.05). SF patients were divided into ~60 (*n* = 34)- and ~80 (*n* = 26)-years-old groups according to age. There was a significant difference in the X-ray diagnosis of mild, moderate or severe SF among patients of different age groups (*χ** = 9.01, *p* < 0.05). Specifically, with increasing age, the number of patients with moderate or severe SF increased (Table 2, Figure 3).

### 3.4. Serum CXCR4 Levels

Quantitative determination of CXCR4 content in the serum of subjects by ELISAs. The median serum CXCR4 levels were significantly higher in the SF group (4.86 ng/mL) than in the control group (4.22 ng/mL) (*Z* = 3.74, *p* < 0.001) (Table 3, Figure 4a).

Under the different results between the SF group and the control group, serum CXCR4 levels were measured in three different aspects, including sex, age and X-ray diagnosis grade. There were no significant differences in serum levels of CXCR4 between males (4.52 ng/mL) and females (5.08 ng/mL) within the SF group (*Z* = 1.68, *p >* 0.05). Within the SF group, the median serum CXCR4 levels were significantly higher in the ~60-year-old subgroup (5.02 ng/mL) than in the ~80-year-old subgroup (4.52 ng/mL) SF (*Z* = 1.95, *p* < 0.05) (Table 4, Figure 4b).

Within the SF group, the median serum CXCR4 levels were significantly different among patients with mild (5.13 ng/mL), moderate (4.29 ng/mL) and severe (4.93 ng/mL) SF (*H* = 12.19, *p* < 0.05; *Z*_mild, moderate_ = 3.43, *p* < 0.001; *Z*_moderate, severe_ = 0.74 and *Z*_mild, severe_ = 0.47, *p* > 0.05). Compared with the moderate SF group, the mild SF group had higher serum CXCR4 levels (Table 4, Figure 4c).

### 3.5. Serum SDF-1 Levels

The median serum SDF-1 levels were significantly higher in the SF group (5.86 ng/mL) than in the control group (5.51 ng/mL) (*Z* = 2.12, *p* < 0.05) (Table 3, Figure 5a).

There were no significant differences in serum levels of SDF-1 between males (5.77 ng/mL) and females (6.07 ng/mL) within the SF group (*Z* = 0.29, *p* > 0.05). Within the SF group, the median serum SDF-1 levels were not significantly different between patients in the ~60-year-old group (5.90 ng/mL) and the ~80-year-old group (5.84 ng/mL) (*Z* = 0.45, *p* > 0.05) (Table 4, Figure 5b). Within the SF group, the median serum SDF-1 levels were not significantly different among patients with mild (5.98 ng/mL), moderate (5.66 ng/mL) and severe (5.52 ng/mL) SF (*H* = 1.95, *p* > 0.05) (Table 4, Figure 5c).

### 3.6. Serum MCP-1 Levels

The median serum MCP-1 levels were significantly higher in the SF group (57.11 pg/mL) than in the control group (48.45 pg/mL) (*Z* = 1.99, *p* < 0.05) (Table 3, Figure 5a).

There were no significant differences in serum levels of MCP-1 between males (54.27 pg/mL) and females (79.34 pg/mL) within the SF group (*Z* = 0.90, *p* > 0.05). Within the SF group, the median serum MCP-1 levels were not significantly different between the ~60-year-old group (80.39 pg/mL) and the ~80-year-old group (47.84 pg/mL) SF (*Z* = 1.22, *p* > 0.05) (Table 4, Figure 6b). Within the SF group, the median serum SDF-1 levels were not significantly different among patients with mild (79.82 pg/mL), moderate (45.11 pg/mL) and severe (12.06 pg/mL) SF (*H* = 4.20, *p* > 0.05) (Table 4, Figure 6c).

### 3.7. Serum ERK1/2 Levels

There were no significant differences in the serum levels of ERK1/2 between the SF group (2.84 ng/mL) and the control group (2.67 ng/mL) (*Z* = 1.07, *p* > 0.05) (Table 3, Figure 7). Analysis of the serum ERK1/2 levels in various aspects is only useful when there are differences in the results between the SF group and the control group. Here, we will not conduct a thorough analysis of the SF group.

### 3.8. Serum NF-κB Levels

The median serum NF-κB levels were significantly higher in the SF group (56.08 ng/mL) than in the control group (53.25 ng/mL) (*Z* = 2.05, *p* < 0.05) (Table 3, Figure 8a).

There were no significant differences in serum levels of NF-κB between males (57.25 ng/mL) and females (54.85 ng/mL) within the SF group (*Z* = 0.75, *p* > 0.05). Within the SF group, the median serum MCP-1 levels were not significantly different between the ~60-year-old group (57.38 ng/mL) and the ~80-year-old group (55.37 ng/mL) SF (*Z* = 0.85, *p* > 0.05) (Table 4, Figure 8b). Within the SF group, the median serum SDF-1 levels were not significantly different among patients with mild (57.58 ng/mL), moderate (54.55 ng/mL) and severe (63.30 ng/mL) SF (*H* = 4.46, *p* > 0.05) (Table 4, Figure 8c).

### 3.9. The Relationships among CXCR4, SDF-1, MCP-1, ERK1/2 and NF-κB in the Serum of SF Patients

In the serum of SF patients, the levels of ERK1/2 and NF-κB were positively correlated (*r* = 0.67, *p* < 0.001) (Figure 9a), and there were positive correlations between the levels of CXCR4 and ERK1/2 (*r* = 0.37, *p* < 0.05) (Figure 9b), CXCR4 and NF-κB *(r* = 0.44, *p* < 0.001) (Figure 9c), SDF-1 and ERK1/2 (*r* = 0.45, *p* < 0.001) (Figure 9d), and SDF-1 and NF-κB (*r* = 0.57, *p* < 0.001) (Figure 9e). There was no correlation between the levels of MCP-1 and ERK1/2 (*r* = 0.02, *p* > 0.05) (Figure 9f) or the levels of MCP-1 and NF-κB (*r* = 0.14, *p* > 0.05) (Figure 9g).

## 4. Discussion

In this study, the X-ray diagnostic grade was first analysed. Finally, the serum levels of SDF-1, CXCR4, MCP-1, ERK1/2 and NF-κB were determined by ELISAs in SF patients, and their correlations were analysed. The results of the current study provide novel evidence of the SDF-1/CXCR4 axis and MCP-1 signalling pathways underlying the occurrence and development of SF.

In this study, there was no statistically significant difference in the sex of patients diagnosed with different grades of SF by X-ray, which was consistent with the previously reported results [40]. However, the X-ray grades of SF were significantly different among patients in different age groups. With increasing age, the X-ray diagnostic grade of SF was more advanced, which may be related to the continuous accumulation of fluorine in the bone over time, resulting in a decrease in the number and proliferative ability of bone marrow-derived mesenchymal stem cells, a reduction in osteogenic force and growth potential [16,41] and an acceleration in changes in bone.

The study found that CXCR4, SDF-1 and MCP-1 may affect the occurrence and sdevelopment of SF by affecting osteocyte metabolism. In the available literature, there are few studies analysing CXCR4, SDF-1 and MCP-1 protein concentrations in SF. To date, CXCR4, SDF-1 and MCP-1 have been analysed in patients with osteoarthritis (OA) and RA. In recent years, studies have found that the SDF-1/CXCR4 axis plays a role in OA, RA and other skeletal diseases [42,43] and is a potential therapeutic target for OA and RA [44]. SF, OA and RA are all bone and joint diseases. Patients with SF had significantly increased serum CXCR4 levels and increased SDF-1 levels, consistent with the previously reported results for RA, OA, OP and others [42,45,46]. Serum MCP-1 levels were also increased in SF patients, which is consistent with research on OA and others [47]. The next step will be to further investigate the mechanism of the CXCR4/SDF-1 axis and MCP-1 in SF through cell experiments to provide a more thorough understanding of the mechanism of SF occurrence and development.

Serum CXCR4 levels were measured in several different aspects. Regarding sex, the study confirmed that serum CXCR4 levels decreased with increasing age. This phenomenon may be associated with the enhancement of OC activity. SF mainly affects osteocytes. With increasing age, bone reconstruction is imbalanced, OC activity is enhanced, and CXCR4 expression in mature OC is downregulated [22]. Regarding X-ray diagnosis grade, this study revealed that serum CXCR4 levels were among patients with mild, moderate or severe SF. Comparative analysis of the groups showed that serum CXCR4 levels were higher in patients with mild SF than in those with moderate SF, suggesting that CXCR4 levels increase with the improvement in X-ray diagnosis grade. The level of CXCR4 can indicate disease severity in the mild and moderate stages. Thus, there is value in further researching CXCR4 in the context of SF.

In this study, serum ERK1/2 levels in the SF group were not different from those in the control group. Interestingly, in contrast to the current study, Jie Gao found that compared with the control group, the fluoride-treated group showed increases in ERK1/2 mRNA and protein levels in OBs in response to different concentrations of fluoride [48]. An article on OA also found that ERK1/2 levels increased in rat experiments [49]. However, ERK1/2 was found to be positively correlated with CXCR4, SDF-1 and NF-κB, suggesting that the CXCR4/SDF-1 axis may affect ERK1/2 activation in SF, which is also affected by RAS, RAF, upstream factors of MEK1/2, MKP-1, tumour necrosis factor-α, etc. [32,50]. The next step will be to increase the amount of serum samples and conduct cell experiments for in-depth research.

In this study, serum NF-κB levels were increased in SF patients, which may be related to the ability of excessive fluoride to promote OC formation and decrease OC apoptosis in the body and thus promote the expression of NF-κB [2,5]. Studies have found that sodium fluoride can upregulate the expression of NF-κB [51]. In the fluoride-induced apoptosis of OCs, NF-κB expression shows a dose-dependent relationship; that is, an increasing fluoride concentration corresponds to an increase in OC apoptosis and a decrease in NF-κB expression [10]. Meanwhile, NF-κB is positively correlated with CXCR4 and SDF-1, indicating that the CXCR4/SDF-1 axis may affect osteocyte synthesis and metabolism by activating the NF-κB pathway in SF.

MCP-1 does not activate the ERK1/2 and NF-κB pathways. It is speculated that other pathways [52], such as the CCL4/CCR5/c-Jun and c-Fos/CCL2 pathways, may affect osteocyte synthesis and metabolism, but the specific mechanism needs to be further studied.

There are several limitations to our study. First, the number of patients was small because we included only patients whose outcomes were definitively identified. However, the patients whose outcomes were definitively identified can make the experimental results more convincing. Second, the number of patients with severe SF was small due to the limited number of patients. It is impossible to present a more detailed and accurate analysis. The next step will be to further investigate the mechanism by which the CXCR4/SDF-1 axis activates the NF-κB pathway in SF and the mechanism of MCP-1 in SF by increasing the number of patients. We require additional analyses at both the protein and gene levels in SF patients to provide a more thorough understanding of the mechanism of SF occurrence and development. Further research is still needed regarding CXCR4 as a marker of SF severity.

## 5. Conclusions

SDF-1 and CXCR4 may activate the NF-κB pathway, and MCP-1 affects the occurrence and development of SF by regulating osteocytes through other pathways.

## Figures and Tables

**Figure 1 ijerph-19-16555-f001:**
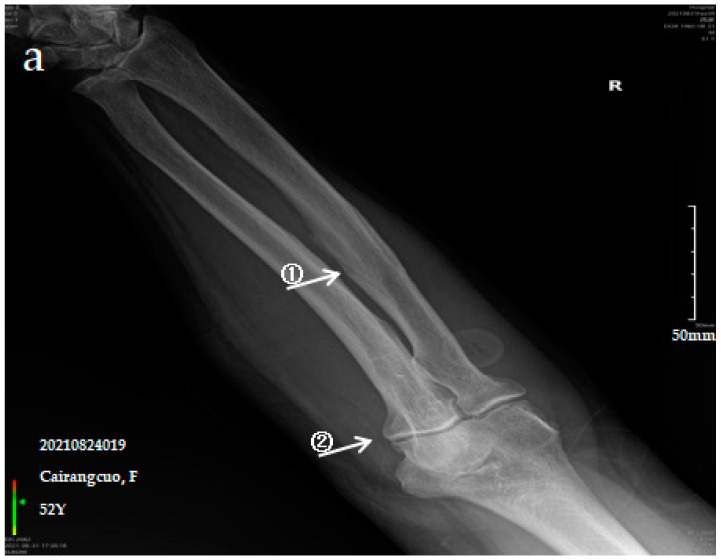
X-ray imaging of SF patients. The anteroposterior actinogram of the upper limb (**a**) and both lower limbs (**b**) of patient with mild SF, the anteroposterior actinogram of the upper limb (**c**) and both lower limbs (**d**) of patient with moderate SF, the anteroposterior actinogram of the upper limb (**e**) and both lower limbs (**f**) of patient with severe SF. **①**: ossification of interosseous membrane; **②**: degeneration of the elbow joint; **③**: Ossification shadow; **④**: Ossification of knee joint capsule; **⑤**: ossification of soleus tendon.

**Figure 2 ijerph-19-16555-f002:**
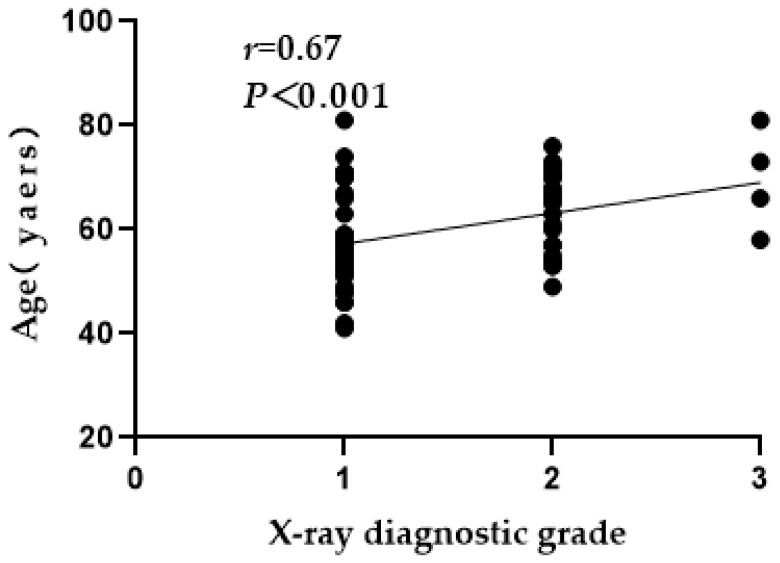
Correlation analysis of X-ray diagnostic grade with age among SF patients. Group 1: mild (*n* = 36), Group 2: moderate (*n* = 20); Group 3: severe (*n* = 4).

**Figure 3 ijerph-19-16555-f003:**
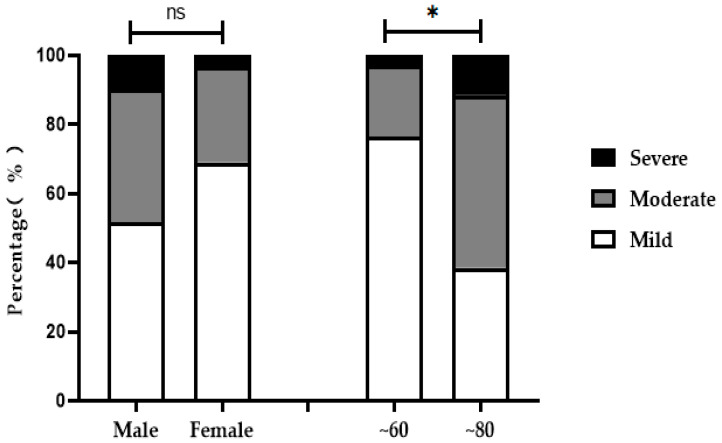
X-ray grade analysis of SF patients of different age groups and sexes. The SF group included 31 males and 29 females and was divided into ~60 (*n* = 34)- and ~80 (*n* = 26)-years-old groups. ^ns^
*p* > 0.05 compared to males; * *p* < 0.05 compared to the ~80-year-old group.

**Figure 4 ijerph-19-16555-f004:**
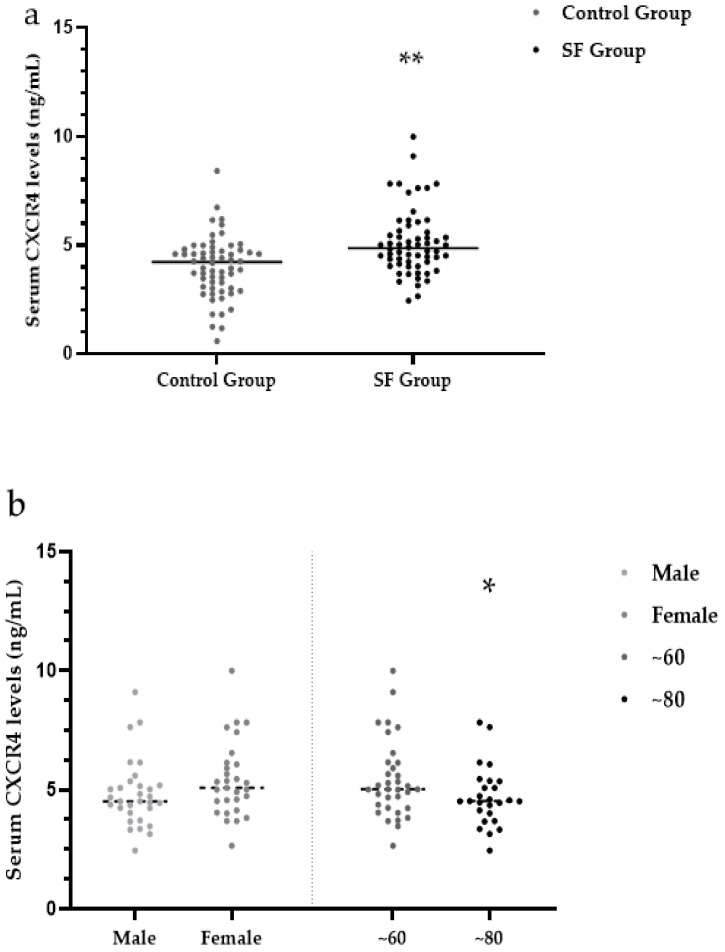
(**a**) Analysis of serum CXCR4 levels in the SF and control groups; (**b**) Analysis of CXCR4 levels in the SF group by sex and age; (**c**) Analysis of CXCR4 levels in the SF group by X-ray diagnostic grade. ** *p <* 0.001 compared to the control group. * *p <* 0.05 compared to ~60. ^##^
*p <* 0.05 compared to both mild and moderate. ^#^
*p <* 0.001 compared to mild. CXCR4: C-X-C motif chemokine receptor 4.

**Figure 5 ijerph-19-16555-f005:**
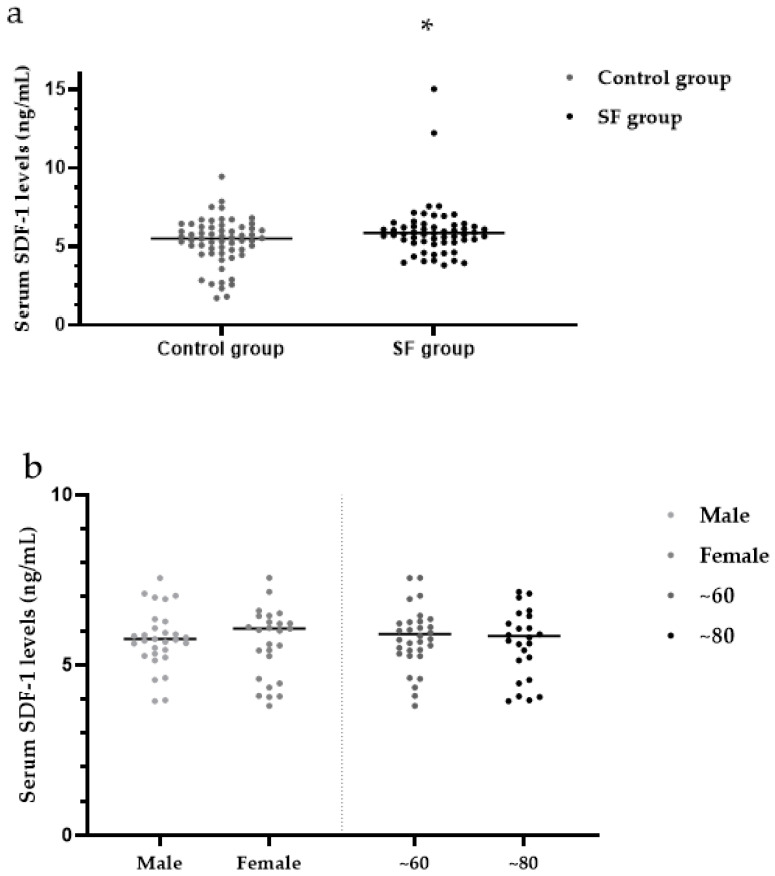
(**a**) Analysis of serum SDF-1 levels in the SF group and control group; (**b**) Analysis of SDF-1 levels in the SF group by sex and age; (**c**) Analysis of SDF-1 levels in the SF group by X-ray diagnostic grade. * *p <* 0.05 compared to the control group. SDF-1: stromal cell-derived factor-1.

**Figure 6 ijerph-19-16555-f006:**
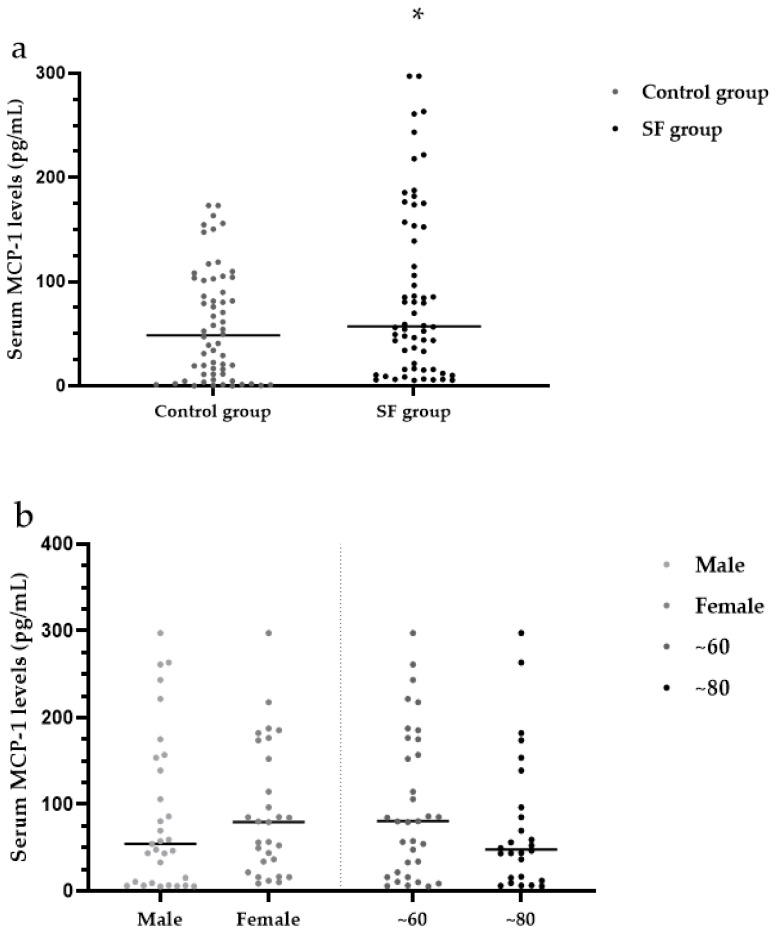
(**a**) Analysis of serum MCP-1 levels in the SF group and control group; (**b**) Analysis of MCP-1 levels in the SF group by sex and age; (**c**) Analysis of MCP-1 levels in the SF group by X-ray diagnostic grade. * *p* < 0.05 compared to the control group. MCP-1: monocyte chemoattractant protein-1.

**Figure 7 ijerph-19-16555-f007:**
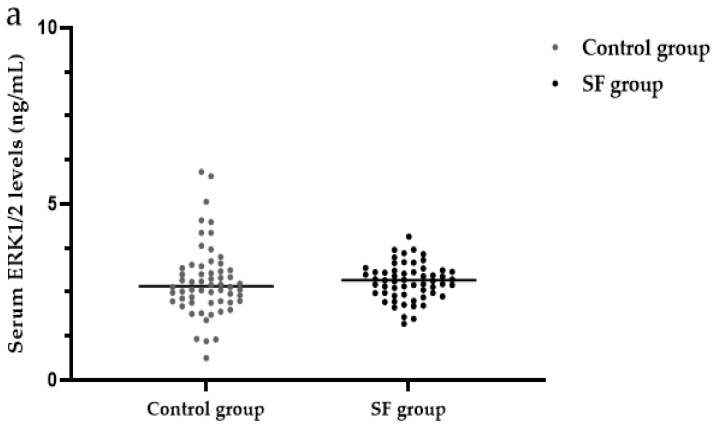
Analysis of serum ERK1/2 levels in the SF group and control group. ERK1/2: extracellular signal-regulated kinase 1/2.

**Figure 8 ijerph-19-16555-f008:**
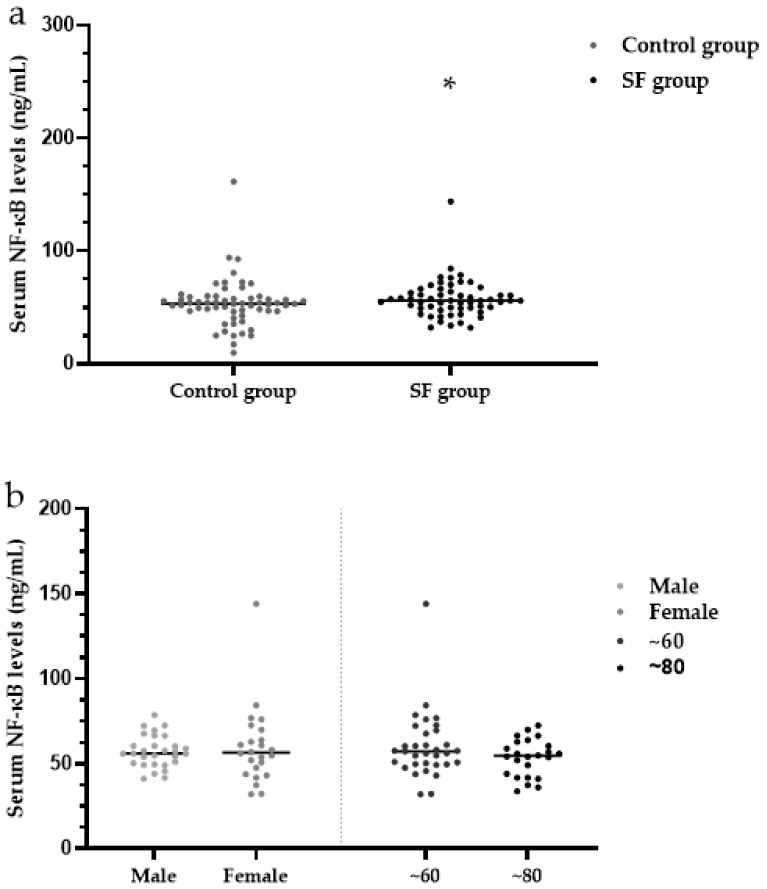
(**a**) Analysis of serum NF-κB levels in the SF group and control group; (**b**) Analysis of NF-κB levels in the SF group by sex and age; (**c**) Analysis of NF-κB levels in the SF group by X-ray diagnostic grade. * *p <* 0.05 compared to the control group. NF-κB: nuclear factor-κB.

**Figure 9 ijerph-19-16555-f009:**
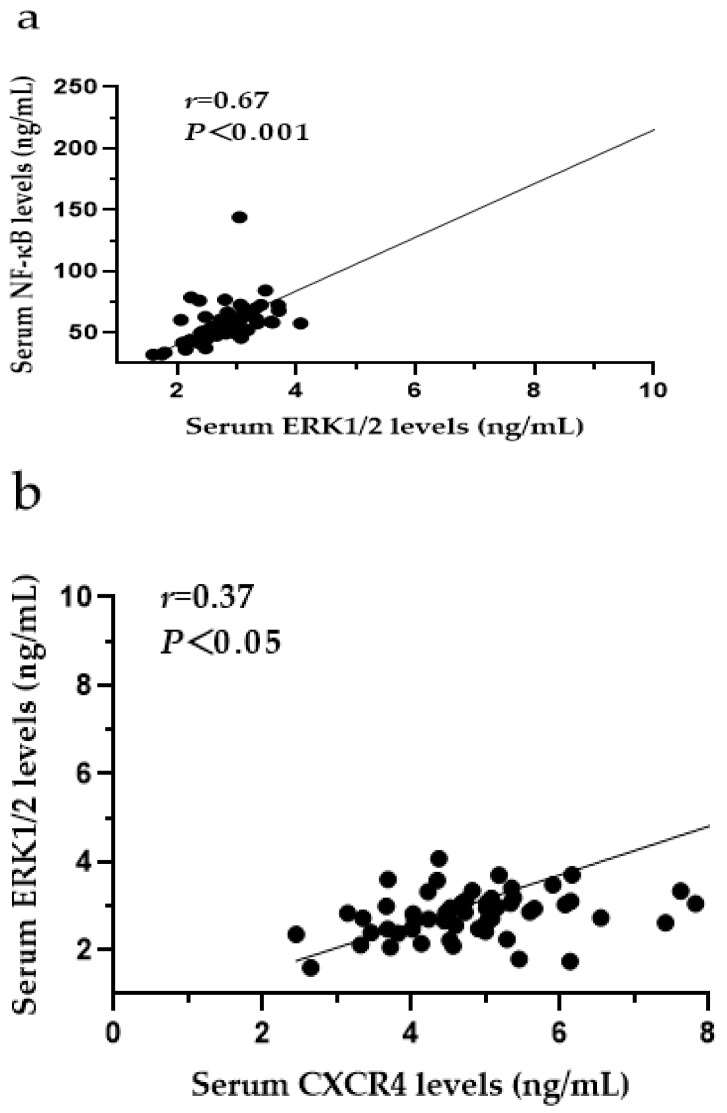
(**a**) Correlation analysis of ERK1/2 and NF-κB in SF (*n* = 60); (**b**) correlation analysis of CXCR4 and ERK1/2; (**c**) correlation analysis of CXCR4 and NF-κB; (**d**) correlation analysis of SDF-1 and ERK1/2; (**e**) correlation analysis of SDF-1 and NF-κB; (**f**) correlation analysis of MCP-1 and ERK1/2; (**g**) correlation analysis of MCP-1 and NF-κB. CXCR4: C-X-C motif chemokine receptor 4; SDF-1: stromal cell-derived factor-1; MCP-1: monocyte chemoattractant protein-1; ERK1/2: extracellular signal-regulated kinase 1/2; NF-κB: nuclear factor-κB.

**Table 1 ijerph-19-16555-t001:** Demographic data.

	SF	Control	Z/*χ** Value	*p*-Value
Age (years)	60.03 ± 9.67	57.42 ± 6.88	1.71	>0.05
Sex (Male/Female)	31/29	28/32	3.00	>0.05
X-ray grade				
Mild	36			
Moderate	20			
Severe	4			

**Table 2 ijerph-19-16555-t002:** X-ray grade analysis of SF patients (*n* = 60) of different ages and sexes.

Characteristic	SF Group N (%)	X-ray Grade			*Χ**	*p*-Value
		Mild	Moderate	Severe		
Sex						
Male	31 (51.67)	16	12	3	2.18	> 0.05
Female	29 (48.33)	20	8	1		
Age (years)						
~60	34 (56.67)	26	7	1	9.01	< 0.05
~80	26 (43.33)	10	13	3		

**Table 3 ijerph-19-16555-t003:** Serum SDF-1, CXCR4, MCP-1, ERK1/2 and NF-κB levels in the SF group and control group [M (*P*_25_, *P*_75_)].

Group	N	CXCR4 (ng/mL)	SDF-1 (ng/mL)	MCP-1 (pg/mL)	ERK1/2 (ng/mL)	NF-κB (ng/mL)
Control group	60	4.22 (3.03, 4.76)	5.51 (4.62, 6.22)	48.45 (11.07, 102.33)	2.67 (2.24, 3.22)	53.25 (46.68, 58.90)
SF group	60	4.86 (4.16, 5.64) **	5.86 (5.28, 6.45) *	57.11 (15.97, 153.30) *	2.84 (2.47, 3.16) **	56.08 (49.29, 66.53) *
*Z* value		3.74	2.12	1.99	1.07	2.05
*p*-value		<0.001	<0.05	<0.05	>0.05	<0.05

** *p <* 0.001, * *p <* 0.05 compared to the control group. CXCR4: C-X-C motif chemokine receptor 4; SDF-1: stromal cell-derived factor-1; MCP-1: monocyte chemoattractant protein-1; ERK1/2: extracellular signal-regulated kinase 1/2; NF-κB: nuclear factor-κB.

**Table 4 ijerph-19-16555-t004:** Serum SDF-1, CXCR4, MCP-1 and NF-κB levels in the SF group (*n* = 60) by sex, age and X-ray diagnosis grade [M (*P*_25_, *P*_75_)].

Characteristic	SF Group N (%)	CXCR4 (ng/mL)	SDF-1 (ng/mL)	MCP-1 (pg/mL)	NF-κB (ng/mL)
Sex					
Male	31 (51.67)	4.52 (4.03, 5.18)	5.77 (5.33, 6.35)	54.27 (9.03, 153.60)	57.25 (50.30, 66.55)
Female	29 (48.33)	5.08 (4.33, 6.10)	6.07 (4.93, 6.49)	79.34 (27.73, 163.15)	54.85 (43.47, 66.98)
Age (years)					
~60	34 (56.67)	5.02 (4.34, 6.14)	5.90 (5.40, 6.38)	80.39 (20.01, 175.48)	57.38 (49.81, 70.30)
~80	26 (43.33)	4.52 (3.93, 5.36)	5.84 (4.99, 6.54)	47.84 (14.29, 107.12)	55.37 (43.67, 64.54)
X-ray grade					
Mild	36 (60.00)	5.13 (4.69, 6.14)	5.98 (5.52, 6.50)	79.82 (38.35, 171.71)	57.58 (50.88, 69.48)
Moderate	20 (33.33)	4.29 (3.67, 4.55)	5.66 (4.71, 6.40)	45.11 (10.49, 136.71)	54.55 (42.28, 59.95)
Severe	4 (6.67)	4.93 (2.96, 7.21)	5.52 (4.28, 12.72)	12.06 (6.66, 118.07)	63.30 (44.89, 126.09)

CXCR4: C-X-C motif chemokine receptor 4; SDF-1: stromal cell-derived factor-1; MCP-1: monocyte chemoattractant protein-1; NF-κB: nuclear factor-κB.

## Data Availability

Not applicable.

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
