# Peer review of "Serum Levels of CXCR4, SDF-1, MCP-1, NF-κB and ERK1/2 in Patients with Skeletal Fluorosis"

_ijerph, 2022, doi:10.3390/ijerph192416555_

Round 1

Reviewer 1 Report

Thank you for your manuscript which presents the CXCR4, SDF-1, MCP-1 NF-KB and ERK1/1 serum levels in patients with skeletal fluorosis. The paper is clear and well written. The item is intriguing and of some practical impact. The authors present that serum SDF-1, CXCR4, MCP-1 and NF-KB levels were significantly higher in the patients with fluorosis than in the control group. Noteworthy, this report has a valuable contribution to the present knowledge of  development mechanism of skeletal fluorosis. The study appears to be properly conducted and the conclusion is validated by the data presented in the manuscript. A couple of point should be better address to enhance the significance of the reported data. In Figure 1a, we can observe two atypical (outliers) results in SF group, significantly greater than the other values. Because atypical observations make it difficult and sometimes even impossible to perform a correct analysis, especially in small groups of patients, therefore, to enhance the significance of the reported data authors should reconsider clinical presentation of these two patients and explanation why such an atypical high result was qualified for further analysis.

Author Response

The atypical high result was qualified for further analysis. I think there are several reasons. First of all, in the selection of SF patients, “Diagnostic Criteria for Endemic Skeletal Fluorosis” (WS 192 -- 2008) were strictly followed, and the interference of diseases such as rheumatoid arthritis and bone and joint diseases was excluded. The clinical manifestations of the two patients were consistent. The second, Although the two atypical (outliers) results are larger than other values, they are within the detection range of the kit. In addition, we have made a parallel comparison of the samples to reduce the random error. Finally, our results are expressed in terms of median, with two atypical (outliers) results having less impact on the overall results.

Reviewer 2 Report

Overall comments:

1.It is noted that your manuscript needs careful editing by someone with expertise in technical English editing paying particular attention to English grammar, spelling, and sentence structure so that the goals and results of the study are clear to the reader.

Abstract:

1.The introduction of the study to the grouping was not scientific enough.It is suggested that 60 patients with SF confirmed by the diagnostic criteria of endemic fluorosis of bone should be taken as SF Group, and 60 healthy people without SF should be taken as control group.

2.By measuring the serum levels of CXCR4, SDF-1, MCP-1, NF-κ B and ERK1/2 in patients with skeletal fluorosis,speculate the possible mechanism, and to provide a new theoretical basis for the occurrence and development of SF.

Introduction:

1.The introduction is too verbose, so it is recommended to make appropriate adjustments to make it more organized.

Materials and Methods:

1.The methods, materials and instruments are suggested to be separated, and the manufacturers and batch numbers are indicated for the materials, and the manufacturers is indicated for the instruments.

Results:

1.Table 1 suggests indicating statistics.

2.In Figure 1, it is recommended to use symbols or arrows to indicate the imaging features in each figure.

3.In Figure 1,the scale of each picture should be consistent with the actual picture size, and it is recommended to confirm again.

4.P values in the text shall be capitalized and italicized.

5.The age groups in Figure 3 should be consistent with Table 2.

6.Table 2 suggests indicating statistics and corresponding P value.

7.The Z value in Table 3 is inconsistent with the description in the text, please confirm.

Discussion:

1.The first paragraph is not very rigorous. In this paper, we measured the levels of four proteins in serum and their correlation, and did not explore the mechanism among the four proteins.

Author Response

Dear Reviewer,

Thank you very much for your valuable comments,

I have revised it according to your comments. I have uploaded the modified article to the webpage.
